# Extractability of Curcuminoids Is Enhanced with Milk and Aqueous-Alcohol Mixtures

**DOI:** 10.3390/molecules27154883

**Published:** 2022-07-30

**Authors:** Raghavendhar R. Kotha, Fakir Shahidullah Tareq, Devanand L. Luthria

**Affiliations:** 1Methods and Application of Food Composition Laboratory, Beltsville Human Nutrition Research Center, Agricultural Research Service, U.S. Department of Agriculture, Beltsville, MD 20705, USA; raghavendhar.kotha@usda.gov (R.R.K.); fakir.tareq@usda.gov (F.S.T.); 2Department of Nutrition and Food Science, College of Agriculture & Natural Resources, 0112 Skinner Building, University of Maryland, College Park, MD 20742, USA

**Keywords:** turmeric, milk, aqueous-alcohol mixtures, curcuminoids, extractability, quantification

## Abstract

In this study, we evaluated the extractability of three curcuminoids (curcumin, demethoxycurcumin, and bisdemethoxycurcumin) from turmeric powder in several solvents using high-performance liquid chromatography (HPLC) with the diode-array detection method. These solvents include water, milk (homogenized, 2% reduced fat, low fat, fat free, soy, almond, coconut, and milkadamia), and aqueous ethanols (0%, 4%, 10%, 20%, 30%, 40%, 50%, and 100%). Ambient water was able to extract only 0.55 mg/g of curcuminoids, whereas warm water extracted more than four-fold higher amounts (2.42 mg/g). Almond, coconut, and milkadamia milk were able to extract only small amounts of curcuminoids at ambient temperatures (0.01–0.07 mg/g). The extractability of curcuminoids in these milk types did not improve, even in warm conditions (0.08–0.37 mg/g). Whereas dairy and soy milk extracted 6.76–9.75 mg/g of curcuminoids under ambient conditions, their extractability increased significantly in warm conditions by 30–100% higher (11.7–14.9 mg/g). The solubility of curcuminoids also varied remarkably in different proportions of aqueous-alcohol mixtures. With 4% ethanol, only 1.7 mg/g of curcuminoids were extracted, and the amounts improved with the increase in ethanol content up to 50% (32.2 mg/g), while 100% ethanol extracted a similar amount as 50% ethanol (34.2 mg/g). This study suggests that the extractability of curcuminoids from turmeric will be dependent on the type of diets consumed with the turmeric supplements.

## 1. Introduction

Turmeric (*Curcuma longa* L.) rhizomes, a plant species belonging to the Zingiberaceae family, have been globally used as a curry spice [1,2]. In Indian ayurvedic medicine, turmeric has been used to treat wound healing, rheumatoid arthritis, urinary tract infections, and liver ailments [1,2]. Turmeric is rich in bioactive polyphenols called curcuminoids (Curcumin (CUR), demethoxycurcumin (DMC), and bisdemethoxycurcumin (BMC)). The health benefits of turmeric are primarily attributed to these three curcuminoids, which account for 1–6% of the total turmeric powder (dry weight) [3]. Both in vivo and human studies suggest that curcumin may act as an anti-inflammatory, anticardiovascular, anticancer, antiaging, and antidiabetic agent [1,4,5].

The therapeutic potential of curcumin, however, is limited by its low solubility in aqueous media, poor bioavailability, and pharmacokinetic profiles [1,3,4,6,7]. To counter these issues, several different formulations (including polymers, lipids, and nanoparticles in appropriate proportions) have been produced and used in multiple studies [8,9,10]. Among the multiple formulation techniques available, using milk as a carrier/media may offer several advantages because milk is widely recognized for its nutrition and health benefits in children and adults. The major proteins in mammalian milk, especially cow and buffalo milk, are casein and whey [11]. Several studies have shown that casein proteins are binders of polyphenols [12,13] and potentially an ideal carrier for the delivery of bioactive molecules such as curcumin and vitamins [14,15]. Moreover, phytochemicals, along with other bioactive molecules already present in milk, may work synergistically and offer additional health benefits. In addition, researchers have already demonstrated the potential delivery of curcumin via caseins to tumor cells and have characterized the interactions between curcumin and casein micelles [14].

The above studies suggest that the efficacy of various nutraceutical products can be enhanced with proper evaluation of the carrier system. We have recently published a review on the biological, pharmaceutical, and nutraceutical aspects of curcuminoids [1]. This was followed by a research article on a simple and rapid extraction method and a sensitive analytical method for the quantification of curcuminoids and common turmeric adulterants using liquid chromatography and tandem mass spectrometry (LC-MS/MS) [16].

With the growing alcoholic beverages demand and consumption, several new alcoholic beverages with various ingredients such as rice, honey, fruits, cassava, pumpkin, herbs, and ginger are currently being developed [17,18,19]. The alcohol content in different liquor products such as beers, wines, vodkas, whiskeys, and rums generally vary between 3 and 50% [20]. These new alcoholic beverages may act as potential delivery carriers for lipophilic bioactive molecules to enhance flavors/aromas, bioavailability, and oxidative stability [21,22]. 

There is a distinct need for a detailed understanding of how the solubility/extractability of curcuminoids in turmeric can be improved using different solvents, such as milk and alcohol. The objective of the current study is to assess different dairy and plant-based milk types for their curcuminoids’ extractability under ambient and warm conditions. In addition, we also investigated the extractability of curcuminoids in eight different aqueous-alcohol solvent mixtures. 

## 2. Results and Discussion

### 2.1. Analytical Method Validation

Calibration curves for three curcuminoids were plotted individually using linear regression analysis of standard concentration vs. peak area, with a 1/x^2^ weighting factor. All three curcuminoids showed excellent linearity with *r*^2^ ≥ 0.99. The linear ranges for all three analytes covered a 5000-fold range of concentration (10–50,000 ng/mL). Retention time, representative linear regression, and r^2^ for CUR, DMC, and BMC were as follows: CUR—retention time 8.6 min, y = 0.1262x − 0.07330, r^2^ = 0.9986; DMC—retention time 8.2 min, y = 0.1394x + 0.007490, r^2^ = 0.9993; BMC—retention time 7.8 min, y = 0.1538x + 0.06625, r^2^ = 0.9998. Table 1 shows the details of accuracy and precision data according to the ICH guidelines [23]. Accuracy (% bias) and precision (RSD) data of all curcuminoids for intra- and interday of all concentration ranges (10, 100, 1000, and 10,000 ng/mL) were in the acceptable range (≤15%). The LOQ for all three curcuminoids was 10 ng/mL.

### 2.2. The Extractability of Curcuminoids in Different Milk Samples

In the present study, we used several different (both dairy and plant-based) commercial milk samples based on their fat and protein content (Table 2). Nutrition data labels showed that the percent fat reported for the milk samples varied 0–10%, and protein content varied from 0 to 8 g. Figure 1 shows the extractability of different curcuminoids per gram of turmeric samples in eight different commercially available milk samples (homogenized, 2% reduced fat, low fat, fat free, soy, almond, coconut, and milkadamia milk) at ambient and warm conditions after warming milk in a microwave for 1 min. The results showed that the extractability of total curcuminoids in ambient conditions varied significantly—over 900-fold—between different milk samples, with fat free showing the maximum (9.75 mg/g) and almond milk the minimum (0.01 mg/g) extractability. Similarly, milkadamia and coconut milk extracted insignificant amounts (≤0.1 mg/g) of curcuminoids. It was interesting to see that there was a moderate decrease (from 9.75 mg/g to 6.76 mg/g) in the extractability of curcuminoids with an increase in fat content from 0% (fat-free milk) to 10% DV (homogenized) milk. This may be attributed to the decreased interaction of curcuminoids to proteins in the presence of lipids, as it is well reported in the literature that curcuminoids bind to casein and whey proteins present in the milk samples [11].

Warming of the milk increased the extractability of curcuminoids significantly (≥30%) in all milk samples. The number of curcuminoids extracted varied by the type of milk selected. An increase of about 80–100% was observed in warming in homogenized milk, 2% reduced-fat milk, and low-fat milk. This can be attributed to the lipophilic nature of curcuminoids. However, a 30–40% increase in curcuminoids content was seen in fat-free and soymilk samples after warming. The extractability of curcuminoids in almond, milkadamia, and coconut milk samples increased about 2- to 8-fold on warming, but the total amounts extracted were still significantly lower (<1%) than the other five milk samples tested in the present study.

### 2.3. The Extractability of Curcuminoids in Different Proportions of Aqueous-Alcohol Mixtures

As reported in our earlier publication, alcohols and acetone are good solvents for the extraction of curcuminoids from turmeric samples [16]. As different alcoholic beverages contain varying percentages of alcohol, ranging from 4–50% [20], we evaluated the extractability of curcuminoids from turmeric in eight different aqueous-alcohol mixtures ranging from 0 to 100% ethanol (0, 4, 10, 20, 30, 40, 50, and 100%). A representative HPLC-DAD chromatogram regarding the curcuminoids extracted from regular and warm homogenized milk, 50% ethanol, and a standard curcuminoids mixture, was shown in Appendix A.

The results presented in Figure 2 showed that the total curcuminoids gradually increased as the proportion of alcohol increased. In water, only 0.55 mg/g of total curcuminoids were extracted. The increase in curcuminoids content was moderate, from 0% to 30% (0% alcohol—0.55 mg/g, 4% alcohol—1.7 mg/g, 10% alcohol—1.95 mg/g, 20% alcohol—3.16 mg/g, 30% alcohol—3.58 mg/g) with a significant increase in all curcuminoids content at 40% alcohol (18.01 mg/g). The extractability of curcuminoids further improved in 50% ethanol (32.22 mg/g), as it was comparable to that of 100% ethanol (34.26 mg/g).

In this study, we also evaluated the solubility of individual curcuminoids to that of total curcuminoids, with various aq. ethanol mixtures (4–100%) The number of individual curcuminoids extracted varied significantly with different aqueous-alcohol mixtures. For instance, the % of CUR, DMC, and BMC extracted in 4% ethanol were 21, 24, and 55%, respectively. Similarly, in 100% ethanol solution, these were 46%, 22%, and 32%, respectively. The DMC% remained constant between 22 and 24% with 10–100% ethanol. Under the same extraction conditions, the amount of CUR gradually increased from 22 to 46%. However, the amount of BMC decreased gradually from 55 to 32% as the ethanol concentration increased from 4 to 100%. These results can be explained due to the varying lipophilic nature of CUR, DMC, and BMC. The Log *P* values for CUR, DMC, and BMC were estimated as 3.62, 3.54, and 3.5, respectively, as documented in the literature (https://pubchem.ncbi.nlm.nih.gov/ (accessed on 26 April 2022).

There have been multiple reviews and research articles reported in the literature to increase the bioavailability of curcuminoids using various formulations [1,24,25,26,27,28]. Maheshwari, in 2010, reported higher solubility of lipophilic curcuminoids in traditional vehicles such as milk, clarified butter, and corn oil, as shown in the current study. Furthermore, the same study also showed a greater permeation of curcuminoids through the intestinal membrane using the non-everted rat intestinal sac model [24]. In a recent study by Han and coworkers, the authors showed that curcumin bioavailability has been improved using different kinds of nano-formulations [25]. For instance, fasting was the best method for the absorption of curcumin in nano-emulsions (Cur-NE) form, and sugar significantly enhanced the bioavailability of curcumin in single-walled carbon nanotubes (Cur-SWNT). However, ingesting milk along with Cur-NE was not recommended. Recently, in a comprehensive review by Yixuan and others, the authors discussed that various curcumin formulations, nano-technology delivery systems, and extraction methods might play a critical role in utilizing the medicinal benefits of curcumin and its derivatives [26]. The practical relevance of the current study is that using warm milk and a very low volume of aqueous-alcohol mixtures can improve the extractability, and potentially the bioefficacy, of curcuminoids.

We evaluated the antioxidant activities of turmeric extracts in eight kinds of milk and eight different aqueous-alcohol mixtures using colorimetric assays (Folin–Ciocalteu (FC) and FRAP) (Appendix A). Good correlations between two colorimetric assays (FC and FRAP) and HPLC analysis of three curcuminoids were observed with different aqueous-alcohol solvents. However, with different milk samples as matrices, the correlations were insignificant (Appendix A), suggesting that it is critical to evaluate antioxidant activity with other complementary chromatography or spectroscopic procedures.

## 3. Materials and Methods

### 3.1. Materials

All chemicals, solvents, milk (homogenized/whole milk (10% DV (daily value) fat), 2% reduced-fat (6% DV fat) milk, low-fat (3% DV fat) milk, fat-free milk, soy milk, almond milk, coconut milk, and milkadamia milk), and turmeric powder were purchased from commercial sources and used without purification. CUR, DMC, and BMC standards were purchased from Sigma-Aldrich Chemical Co. (St. Louis, MO, USA). Deionized water (18 Ω) was obtained using a Millipore Milli-Q purification system (New Bedford, MA, USA). LC−MS grade acetonitrile and water were purchased from Fischer Scientific (Pittsburg, PA, USA). HPLC grade pure ethanol 200-proof was purchased from Sigma-Aldrich Chemical Co. (St. Louis, MO, USA). Table 2 contains the nutrient label data on the milk samples. Extractions were carried out in 15 mL disposable polypropylene centrifuge tubes (Fisher Scientific, Pittsburgh, PA, USA). Poly (vinylidene difluoride) (PVDF) syringe filters with a pore size of 0.45 μm were purchased from National Scientific (Duluth, GA, USA). A Rival 700-Watt Microwave oven was used for warming the water and milk samples.

### 3.2. Preparation of Standard and Sample Solutions

The individual stock solutions of CUR, DMC, and BMC were prepared with 1 mg/mL concentrations. All analytes were individually weighed accurately with an analytical balance and dissolved in methanol. All three standards were combined into equal proportions to make a final concentration of each analyte, 100 μg/mL, as it was used as the primary stock solution. Calibration curves were constructed in the range of 10−50,000 ng/mL (10, 50, 100, 500, 1000, 5000, 10,000, and 50,000 ng/mL) for all three analytes. The quality control (QC) samples were prepared in the same way as above, with concentrations of 10, 100, 1000, and 10,000 ng/mL for CUR, DMC, and BMC.

### 3.3. Sample Extraction

Curcuminoids were extracted from turmeric powder using a previously reported optimized ultrasonic extraction method [16]. Briefly, turmeric samples (100 ± 0.5 mg) were placed into disposable 15 mL centrifuge tubes and extracted with 5 mL of different kinds of ambient and warm milks (70–80 °C) and various aqueous ethanol (ambient) solvent mixtures (ambient temperature) in triplicates. All samples were vortexed for about 15 s and extracted for 5 min using ultrasound-assisted extraction. All extracts were centrifuged for 10 min at 4000× *g* and filtered through a 0.45 μm PVDF syringe filter prior to the analysis. For milk extracts, 100 µL of the filtered extract was added to 900 µL of pure ethanol for protein precipitation.

### 3.4. Instrumentation

All samples were analyzed using an Agilent 1290 HPLC with a UV diode-array detector. Chromatographic separations were carried out using an Agilent Eclipse Plus C18 (1.8 μm, 4.6 × 50 mm) column equipped with an Agilent guard column. The column was maintained at 35 °C with a flow rate of 0.7 mL/min. The mobile phases used for the separations: solvent A (water with 0.1% formic acid) and solvent B (acetonitrile with 0.1% formic acid). The sample injection volume was 10 μL. The total chromatography run time was 15 min. The solvent gradient was 0.0–3.0 min 30–40% B, 3.0–10 min 40–60% B, 10.0–10.3 min 60–95% B, 10.3–13.3 min 95% B, 13.3–13.5 min 95–30% B, and 13.5–15.0 min 30% B. UV detector wavelength 425 nm was used for the quantification of curcuminoids.

### 3.5. Method Validation and Statistical Analysis

All calculations were based on the peak area response of the analyte. Data were collected and processed with Agilent ChemStation. All data analyses and statistics were performed using Microsoft Excel. Linearity was measured by calibration curves plotted from 10 non-zero standards to evaluate the working analytical linearity range. Samples’ concentrations were determined by linear regression analysis of a plot of calibration standard concentration vs. the peak area of standard with a 1/x^2^ weighting factor. The lowest point on the calibration curve was used as the low limit of quantification (LOQ). Accuracy and precision were determined using triplicates of four different concentrations (10, 100, 1000, and 10,000 ng/mL) on two different days for intra- and inter-day analysis. Accuracy and precision were measured as the % of calculated vs. theoretical concentrations of the sample, and the relative standard deviation (RSD), respectively. One-way ANOVA was used to compare extractability of curcuminoids with different milk samples using the JMP Pro 15.0.0 Statistical software (Cary, NC, USA). The mean comparison for all pairs was made with Tukey–Kramer HSD.

## 4. Conclusions

In conclusion, the extractability of curcuminoids varied significantly with the type of milk and aqueous-alcohol mixtures. Both dairy and soy milk showed significant extractability of curcuminoids from turmeric compared to water. However, almond, coconut, and milkadamia milk only extracted very small amounts of curcuminoids at ambient temperatures. Optimum extractions of curcuminoids were obtained with 40%, 50% and 100% ethanol–water solvent mixtures. Warming milk and water also significantly improved the extractability of curcuminoids. Good correlations for the determination of the curcuminoids content between two colorimetric assays (FC and FRAP) and HPLC analysis were observed with different aqueous-alcohol solvents, but with different milk, the correlations were significantly reduced. This study shows that the bioavailability and bioefficacy of phytochemicals from various foods or dietary supplements significantly change with the mode of consumption.

## Figures and Tables

**Figure 1 molecules-27-04883-f001:**
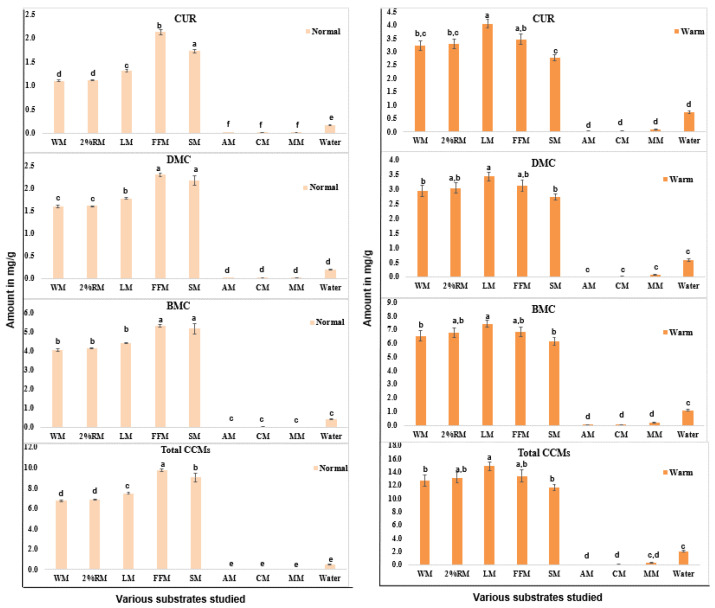
The extractability of bioactive curcuminoids, curcumin (CUR), demethoxycurcumin (DMC), bisdemethoxycurcumin (BMC), and total curcuminoids (CCMs) in different dairy and plant-based milk (WM—homogenized, 2%RM—2% reduced-fat, LM—low-fat, FFM—fat-free, SM—soy, AM—almond, CM—coconut, and MM—milkadamia) at room temperature (**left**) and warm milk (**right**). Water was used as a control. The error bars denote the standard deviation of triplicate analyses. Levels not represented by the same letter (a, b, c, d, e, or f) are significantly different based on one-way ANOVA.

**Figure 2 molecules-27-04883-f002:**
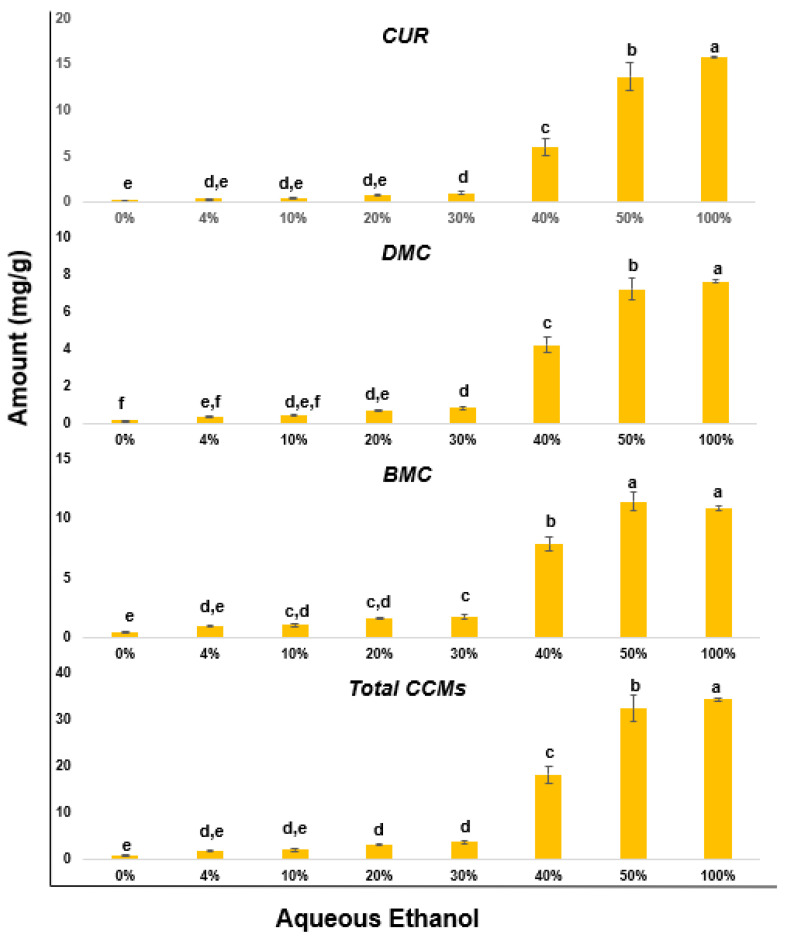
Extractability of three bioactive curcuminoids, curcumin (CUR), demethoxycurcumin (DMC), and bisdemethoxycurcumin (BMC) in different aqueous-alcohol solutions (0, 4, 10, 20, 30, 40, 50, and 100% aq. alcohols). The error bars denote the standard deviation of triplicates. Levels not represented by the same letter (a, b, c, d, e, or f) are significantly different based on one-way ANOVA.

**Table 1 molecules-27-04883-t001:** Accuracy and precision data for the three curcuminoids (curcumin (CUR), demethoxycurcumin (DMC), and bisdemethoxycurcumin (BMC)) were determined by the HPLC diode-array detection method.

Analyte	QC Level (ng/mL)	Intra-Day (*n* = 3)			Interday (*n* = 6; 2 × 3)		
		Mean ± SD (ng/mL)	Accuracy %	RSD %	Mean ± SD (ng/mL)	Accuracy %	RSD %
**Curcumin (CUR)**	10	10.62 ± 0.37	106.18	3.52	9.64 ± 1.13	96.43	11.74
	100	92.55 ± 6.23	92.55	6.74	92.09 ± 4.72	92.10	5.13
	1000	976.17 ± 8.22	97.62	0.84	982.15 ± 12.58	98.20	1.28
	10,000	10,357.27 ± 466.06	103.57	4.50	10,217.33 ± 403.43	102.20	3.95
**Demethoxycurcumin (DMC)**	10	9.61 ± 0.34	96.12	3.57	9.49 ± 0.53	94.89	5.63
	100	96.61 ± 3.12	96.61	3.23	95.47 ± 4.89	95.50	5.12
	1000	991.69 ± 4.09	99.17	0.41	992.26 ± 18.71	99.20	1.89
	10,000	10,147.24 ± 264.25	101.47	2.60	10,082.29 ± 296.44	100.80	2.94
**bisDemethoxycurcumin (BMC)**	10	9.76 ± 0.81	97.56	8.31	9.70 ± 0.79	96.97	8.19
	100	101.20 ± 3.55	101.20	3.51	101.27 ± 2.62	101.30	2.59
	1000	1007.07 ± 24.41	100.71	2.42	995.65 ± 20.92	99.57	2.10
	10,000	10,253.55 ± 185.62	102.54	1.81	100,025.44 ± 263.53	100.25	2.63

**Table 2 molecules-27-04883-t002:** Amounts of fat and protein per serving in the nutrient data labels provided with different milk samples.

Milk	Amount of Fat (g)	Concentration (mg/mL) *	Fat (% DV)	Amount of Protein (g)	Concentration (mg/mL) *
Homogenized	8	33.3	10	8	33.3
2% Reduced fat	5	20.8	8	8	33.3
Low fat	2.5	10.4	4	8	33.3
Fat free	0	0.0	0	8	33.3
Soy	4.5	18.8	6	8	33.3
Almond	2.5	10.4	3	1	4.2
Coconut	4	16.7	5	0	0.0
Milkadamia	3.5	14.6	4	1	4.2

* Concentrations were determined using serving size (240 mL) as volume.

## Data Availability

Not applicable.

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
