# Peer review of "Extractability of Curcuminoids Is Enhanced with Milk and Aqueous-Alcohol Mixtures"

_molecules, 2022, doi:10.3390/molecules27154883_

Round 1

Reviewer 1 Report

Summary:

This article describes the comparison of extraction solvents on the extraction of curcuminoids from turmeric. The authors have previously published the optimized analytical method, therefore as I understand it, the impetus for this study is to investigate potential liquid carrier systems to provide the health effects of curcuminoids when ingested as a drink. Presumably, when ingested in solid form, the bioavailability of these compounds will be controlled by body fluids such as saliva and stomach acid. The authors should expand on this in the introduction in order to provide a rationale for this study. Why is a liquid carrier system preferred compared to digesting as a solid? Is the bioavailability very low when ingested as a solid? Also, I question if the negative health effects of ethanol consumption might outweigh any health benefits associated with ingesting curcuminoids.

The supplementary information appears to have little relevance to the manuscript.  Phenolic content and antioxidant activity are barely mentioned in the manuscript. The authors need to better explain the relevance of these measurements or remove them entirely. Table S1 does not appear in the manuscript or supplemental information.

I do not understand the data presented in the second to last paragraph on page 5. Figure 2 clearly show that the amount of all the curcuminoids extracted increases with increasing ethanol concentration. Yet, according to the text, the percent extraction does not necessarily follow the same trend. Indeed, “BMC decreases gradually from 55% to 32% as the ethanol concentration was increased”.  Why the difference between amount extracted and extraction efficient? How is the extraction efficiency calculated?  I don’t follow the explanation this is due to the varying lipophilic natures.

The manuscript is generally well written with appropriate statistical data treatment.

Minor corrections:

-Figure 2 is confusing. The x-axis is %ethanol for each graph. Please use a different size and type of font for the graph titles so that they are easily differentiated for the x-axis label.

-I’m not familiar with a 1/x2 weighting factor. Please explain what it is and why it was used.

-Include the abbreviation CUR after curcumin on the 5th line of the introduction.

-Please include a representative labelled chromatogram.

-Table S1 does not appear in the manuscript or supporting information.

Author Response

This article describes the comparison of extraction solvents on the extraction of curcuminoids from turmeric. The authors have previously published the optimized analytical method, therefore as I understand it, the impetus for this study is to investigate potential liquid carrier systems to provide the health effects of curcuminoids when ingested as a drink. Presumably, when ingested in solid form, the bioavailability of these compounds will be controlled by body fluids such as saliva and stomach acid. The authors should expand on this in the introduction in order to provide a rationale for this study. Why is a liquid carrier system preferred compared to digesting as a solid? Is the bioavailability very low when ingested as a solid?

Response: Curcumin is yet to reach the status of a therapeutic drug candidate mainly because a standard solid dosage of curcumin suffers from poor oral bioavailability (0.05 μg mL−1), less than 1%) (Reference- Suresh, K.; Ashwini Nangi, A. Curcumin: pharmaceutical solids as a platform to improve solubility and bioavailability. CrystEngComm, 2018,20, 3277-3296 https://doi.org/10.1039/C8CE00469B)

Moreover, several different formulations (materials/ mixtures that combined curcumin with other elements, including polymers, lipids, and nanoparticles in appropriate proportions) have been produced and used in multiple studies to increase the bioavailability and pharmacokinetic profiles (Reference- Kotha, R. R.; Luthria, D. L. Curcumin: Biological, Pharmaceutical, Nutraceutical, and Analytical Aspects. Molecules 2019, 24 (16), 2930. https://doi.org/10.3390/molecules24162930.).

Also, I question if the negative health effects of ethanol consumption might outweigh any health benefits associated with ingesting curcuminoids.

Response:  I agree with the reviewer’s comment about the negative health effects of alcohol consumption.  We are not recommending the consumption of large volumes of alcoholic beverages, but we are showing the solubility is significantly enhanced with very small amounts of alcohol usage.  Furthermore, the objective of the study is to show the importance of how diets can influence the extractability of curcuminoids and other bioactive compounds present in food and dietary supplement.

The supplementary information appears to have little relevance to the manuscript. Phenolic content and antioxidant activity are barely mentioned in the manuscript. The authors need to better explain the relevance of these measurements or remove them entirely.

Response: There have been several reports on issues related to the efficacy of the colorimetric and other antioxidant methods for the assay of antioxidant activity (References: 1. Singleton, V. L.; Orthofer, R.; Lamuela-Raventós, R. M. [14] Analysis of total phenols and other oxidation substrates and antioxidants by means of Folin-Ciocalteu reagent. In Methods in Enzymology; Academic Press, 1999; Vol. 299, pp 152−178; 2. Tareq, F. S.; Kotha, R. R.; Ferreira, J. F. S.; Sandhu, D.; Luthria, D. L. Influence of Moderate to High Salinity on the Phytochemical Profiles of Two Salinity-Tolerant Spinach Genotypes. ACS Food Sci. Technol. 2021, 1 (2), 205–214. https://doi.org/10.1021/acsfoodscitech.0c00034.  This study further supports the need for detailed chromatographic analysis methods besides colorimetric antioxidant methodology (Folin ciocalteu and ferric reducing antioxidant power) to confirm total antioxidant activity.  Hence, we have added the antioxidant activity results in the supporting information and summarized the results in one paragraph in the manuscript.  But if the reviewer strongly feels that this information needs to be deleted, we will be glad to do it.

 Table S1 does not appear in the manuscript or supplemental information.

Response: Thank you for finding the typo. We changed Table S1 to Table 2.

I do not understand the data presented in the second to last paragraph on page 5. Figure 2 clearly shows that the amount of all the curcuminoids extracted increases with increasing ethanol concentration. Yet, according to the text, the percent extraction does not necessarily follow the same trend. Indeed, “BMC decreases gradually from 55% to 32% as the ethanol concentration was increased”. Why the difference between amount extracted and extraction efficient? How is the extraction efficiency calculated? I don’t follow the explanation this is due to the varying lipophilic natures.
Response: Thank you for your comment. Here, we tried to compare the % of each individual curcuminoid to that of total curcuminoids for each aq. alcohol (from 10-100%) solvents. For instance, the % of CUR, DMC, and BMC extracted in 4% ethanol were 21, 24, and 55, respectively. Similarly, in 100% ethanol solution these were 46%, 22%, and 32%, respectively.

We have modified the paragraph as suggested. Furthermore, the extraction efficiency was calculated as either individual or combined curcuminoids extracted with a defined extraction solvent (different compositions of aq. alcohol mixtures).

The manuscript is generally well written with appropriate statistical data treatment.

Response: Thank you

Minor corrections:

-Figure 2 is confusing. The x-axis is %ethanol for each graph. Please use a different size and type of font for the graph titles so that they are easily differentiated for the x-axis label.

Response: We modified the Figure 2, as suggested.

-I’m not familiar with a 1/x2 weighting factor. Please explain what it is and why it was used.

Response: Please refer to the below reference mentioned that weighting factor 1/x2 should always be used as the weighting factor for all bioanalytical LC-MS/MS assays.

Reference: Huidong Gu*, Guowen Liu, Jian Wang, Anne-Françoise Aubry, and Mark E. Arnold. Selecting the Correct Weighting Factors for Linear and Quadratic Calibration Curves with Least-Squares Regression Algorithm in Bioanalytical LC-MS/MS Assays and Impacts of Using Incorrect Weighting Factors on Curve Stability, Data Quality, and Assay Performance. Anal. Chem. 2014, 86, 18, 8959–8966. https://pubs.acs.org/doi/pdf/10.1021/ac5018265

A paragraph from the abstract of above reference copied here: “The correct weighting factor is determined by the relationship between the standard deviation of instrument responses (σ) and the concentrations (x). The weighting factor of 1, 1/x, or 1/x2 should be selected if, over the entire concentration range, σ is a constant, σ2 is proportional to x, or σ is proportional to x, respectively. For the first time, we demonstrated with detailed scientific reasoning, solid historical data, and convincing justification that 1/x2 should always be used as the weighting factor for all bioanalytical LC-MS/MS assays.”

Moreover, it is our and other researchers’ observation that weighting factor should be used for calibration curve, especially for wide linearity range.

-Include the abbreviation CUR after curcumin on the 5th line of the introduction.

Response: included as suggested.

-Please include a representative labelled chromatogram.

Response: As suggested, we included a representative chromatogram (Figure S1) and added a relevant sentence in Section 2.3.

-Table S1 does not appear in the manuscript or supporting information.

Response: Changed Table S1 to Table 2 as suggested.

Reviewer 2 Report

The study aims to describe the influence of possible food matrices on the extraction of curcuminoids. Different kinds of milk and milk substituents as well as water-ethanol mixtures were used. The highest amounts of total curcuminoids were found to be extracted with high percentage ethanol, followed by warm dairy milk.

My main concerns are related to method validation and data analysis procedures:

1. With regard to accuracy determination, the ICH methodology you cite recommends either analyzing reference material or comparing the results with those of another validated method. You, if I understand well, only compare the measured concentrations with the expected ones. In my opinion, this is not sufficient to determine the accuracy.

2. Please specify your criteria for the determination of LOQ.

3. What temperature did the milk have after warming in a microwave oven (Chapter 2.2)?

4. When comparing more than two groups of samples, the Student’s t test cannot be used (Figures 1 and 2). I recommend using ANOVA or a similar test designated for the comparison of multiple groups.

5. In the presented form, the axes labels in Figure 1 are not easily readable.

6. Please replace amounts with concentrations in Table 2.

In addition, the study is not well connected with recent progress made in this area.

7. Recent studies should be mentioned (10.1002/jsfa.11215 and several articles reviewed, e.g. in 10.1016/j.indcrop.2021.114050, 10.3390/pharmaceutics13122102, or 10.3390/pharmaceutics13101715).

8. The contribution and practical relevance of the study are not very clearly described. Can bioavailability be inferred from the extraction data? (Please compare your results to 10.1016/j/jff.2010.01.004)

Depending on the maximal acceptable length of the Communication paper, a more in-depth discussion is preferable:

7. What could be the possible mechanism by which fat reduces protein-curcuminoid interactions?

8. Similarly, what compounds present in milk could influence the results of FC and FRAP?

Other comments:

9. The following paragraph is redundant: ‘The above studies suggest that the efficacy of various nutraceutical products can be enhanced with proper evaluation of the carrier system. We have recently published a review on the biological, pharmaceutical, and nutraceutical aspects of curcuminoids.1 This was followed by a research article on a rapid and sensitive analytical method for the quantification of curcuminoids and common turmeric adulterants using liquid chromatography and tandem mass spectrometry (LC-MS/MS). 16 In the same published manuscript, we also reported a simple rapid extraction methodology using alcohol as an extraction solvent.’

10. Can alcoholic beverages be considered nutraceuticals with a clear conscience?

11. You use the expressions ‘milkamedia’ and ‘milkademia’ throughout the text. Do you mean ‘milkadamia’?

Author Response

Comments and Suggestions for Authors
The study aims to describe the influence of possible food matrices on the extraction of curcuminoids. Different kinds of milk and milk substituents as well as water-ethanol mixtures were used. The highest amounts of total curcuminoids were found to be extracted with high percentage of ethanol, followed by warm dairy milk.

My main concerns are related to method validation and data analysis procedures:

1. With regard to accuracy determination, the ICH methodology you cite recommends either analyzing reference material or comparing the results with those of another validated method. You, if I understand well, only compare the measured concentrations with the expected ones. In my opinion, this is not sufficient to determine the accuracy.

Response: Thank you for your comment.

As per ICH or FDA guidelines, Accuracy, often measured as %bias is calculated by the below formula;

We believe the Accuracy measurement using above formula is widely accepted in the research community as well as recommended by FDA and ICH. We had added few recent references from the scientific literature.

References:

  1. FDA, Bioanalytical Method Validation Guidance for Industry

https://www.fda.gov/media/70858/download

  1. ICH, Validation of Analytical Procedures: Text and Methodology q2(r1)

  1. L. Yin, T. Wang, M. Shi, Y. Zhang, X. Zhao, Y. Yang, J. Gu, Simultaneousdetermination of ten antiepileptic drugs in human plasma by liquidchromatography and tandem mass spectrometry with positive/negativeion-switching electrospray ionization and its application in therapeutic drugmonitoring, J. Sep. Sci. 39 (2016) 964–972. https://doi.org/10.1002/jssc.201501067
  2. M. Shibata, S. Hashi, H. Nakanishi, S. Masuda, T. Katsura, I. Yano, Detection of22 antiepileptic drugs by ultra-performance liquid chromatography coupledwith tandem mass spectrometry applicable to routine therapeutic drugmonitoring, Biomed. Chromatogr. 26 (2012) 1519–1528. https://doi.org/10.1002/bmc.2726
  3. Tian Liu, Raghavendhar R. Kotha, Jace W. Jones, James E. Polli, Maureen A. Kane, Fast liquid chromatography-tandem mass spectrometry method for simultaneous determination of eight antiepileptic drugs and an active metabolite in human plasma using polarity switching and timed selected reaction monitoring, Journal of Pharmaceutical and Biomedical Analysis. 2019, 176, 112816. https://doi.org/10.1016/j.jpba.2019.112816.

  1. Please specify your criteria for the determination of LOQ.

Response: Thank you for your comment.

We used the lowest point on the calibration curve was used as the low limit of quantification (LOQ) as described in Section 3.5

3. What temperature did the milk have after warming in a microwave oven (Chapter 2.2)?

Response

The temperature of the milk samples varied between 70-80oC and we have added this information in the manuscript (Pg #10, line # 214). Moreover, the objective of the study is to evaluate the impact of the usage of warm milk which is commonly consumed in India and other Asian countries as compared to cold milk beverages preferred in the western world.

  1. When comparing more than two groups of samples, the Student’s t test cannot be used (Figures 1 and 2). I recommend using ANOVA or a similar test designated for the comparison of multiple groups.

As suggested, we have carried out the one-way ANOVA analysis for comparison of extractability of curcuminoids between different samples using the JMP Pro 15.0.0 Statistical software (Cary, NC). The mean comparison for all pairs was made with Tukey-Kramer HSD test. The related information in the manuscript has been modified.

5. In the presented form, the axes labels in Figure 1 are not easily readable.
Response: As suggested, we have changed the font size bigger, so that the axis labels are easily legible.

  1. Please replace amounts with concentrations in Table 2.
    Response: As suggested we have incorporated the needed changes.

Modified the Table 2 as suggested.

In addition, the study is not well connected with recent progress made in this area.

 Recent studies should be mentioned (10.1002/jsfa.11215 and several articles reviewed, e.g. in 10.1016/j.indcrop.2021.114050, 10.3390/pharmaceutics13122102, or 10.3390/pharmaceutics13101715).

8. The contribution and practical relevance of the study are not very clearly described. Can bioavailability be inferred from the extraction data? (Please compare your results to 10.1016/j/jff.2010.01.004). Depending on the maximal acceptable length of the Communication paper, a more in-depth discussion is preferable:

Response: As suggested, we have added additional references including reviews and related discussion in the revised manuscript. (Page # 7-8 and line numbers 160-177). Also, we added the references 24-28 in the references section.
7. What could be the possible mechanism by which fat reduces protein-curcuminoid interactions?

Response: As suggested, it can be a topic for further studies that is beyond the scope of the current practical application.

  1. Similarly, what compounds present in milk could influence the results of FC and FRAP?
    Response: As suggested, it can be a topic for further studies that are beyond the scope of the current practical application. Both, Folin ciocalteu and FRAP methods have been extensively used in the literature. We suggest these spectroscopic methods have limitations and the results need to be confirmed with detailed chromatographic methods.

We have earlier reported that carbohydrates interfere with colorimetric assays like FC and FRAP. Moreover, other molecules in milks, have been reported to have antioxidant activity (References: 1. Zulueta, A.; Maurizi, A.; Frígola, A.; Esteve, M. J.; Coli, R.; Burini, G., Antioxidant capacity of cow milk, whey and deproteinized milk. International Dairy Journal 2009, 19 (6), 380-385. https://doi.org/10.1016/j.idairyj.2009.02.003. 2. Cloetens, L.; Panee, J.; Åkesson, B., The antioxidant capacity of milk--the application of different methods in vitro and in vivo. Cellular and molecular biology (Noisy-le-Grand, France) 2013, 59 (1), 43-57)

Other comments:

9. The following paragraph is redundant: ‘The above studies suggest that the efficacy of various nutraceutical products can be enhanced with proper evaluation of the carrier system. We have recently published a review on the biological, pharmaceutical, and nutraceutical aspects of curcuminoids.1 This was followed by a research article on a rapid and sensitive analytical method for the quantification of curcuminoids and common turmeric adulterants using liquid chromatography and tandem mass spectrometry (LC-MS/MS). 16 In the same published manuscript, we also reported a simple rapid extraction methodology using alcohol as an extraction solvent.’

Response: We agree with the reviewer’s comment and have modified the paragraph, as suggested and deleted the final repeated sentence.

10. Can alcoholic beverages be considered nutraceuticals with a clear conscience?

Response: We are not considering alcoholic beverages as nutraceutical products but there have been multiple reports on alcoholic beverages as potential nutraceuticals. Please refer to related references.

References 21 and 22 from the manuscript.

(21)     Ducruet, J.; Rébénaque, P.; Diserens, S.; Kosińska-Cagnazzo, A.; Héritier, I.; Andlauer, W. Amber Ale Beer Enriched with Goji Berries – The Effect on Bioactive Compound Content and Sensorial Properties. Food Chem. 2017, 226, 109–118. https://doi.org/10.1016/j.foodchem.2017.01.047.

(22)     Di Renzo, L.; Cioccoloni, G.; Sinibaldi Salimei, P.; Ceravolo, I.; De Lorenzo, A.; Gratteri, S. Alcoholic Beverage and Meal Choices for the Prevention of Noncommunicable Diseases: A Randomized Nutrigenomic Trial. Oxid. Med. Cell. Longev. 2018, 2018, 5461436. https://doi.org/10.1155/2018/5461436.

11. You use the expressions ‘milkamedia’ and ‘milkademia’ throughout the text. Do you mean ‘milkadamia’?

Response: Thank you for your observation. We have changed it to ‘milkadamia’ throughout the manuscript.